# Probe-Based Real-Time qPCR Assays for a Reliable Differentiation of Capripox Virus Species

**DOI:** 10.3390/microorganisms9040765

**Published:** 2021-04-06

**Authors:** Janika Wolff, Martin Beer, Bernd Hoffmann

**Affiliations:** Institute of Diagnostic Virology, Friedrich-Loeffler-Institut, Federal Research Institute for Animal Health, Südufer 10, D-17493 Greifswald-Insel Riems, Germany; Janika.Wolff@fli.de (J.W.); Martin.Beer@fli.de (M.B.)

**Keywords:** capripox, lumpy skin disease, goatpox, sheeppox, LSDV, GTPV, SPPV, species-specific, TaqMan, MGB

## Abstract

Outbreaks of the three capripox virus species, namely lumpy skin disease virus, sheeppox virus, and goatpox virus, severely affect animal health and both national and international economies. Therefore, the World Organization for Animal Health (OIE) classified them as notifiable diseases. Until now, discrimination of capripox virus species was possible by using different conventional PCR protocols. However, more sophisticated probe-based real-time qPCR systems addressing this issue are, to our knowledge, still missing. In the present study, we developed several duplex qPCR assays consisting of different types of fluorescence-labelled probes that are highly sensitive and show a high analytical specificity. Finally, our assays were combined with already published diagnostic methods to a diagnostic workflow that enables time-saving, reliable, and robust detection, differentiation, and characterization of capripox virus isolates.

## 1. Introduction

The three species lumpy skin disease virus, goatpox virus, and sheeppox virus form the genus *Capripoxvirus* within the family *Poxviridae* [1]. Capripox viruses (CaPVs), mainly infecting cattle, goats, and sheep, respectively [2,3], are described as the most serious poxvirus diseases of domestic animals [2,4,5,6]. Due to severe production losses caused by CaPV outbreaks (e.g., decreased growth rate and mass loss, decreased milk yield, damage to hide and skin as well as temporary or permanent infertility in bulls), outbreaks of these diseases have a massive impact on national as well as global economies [2,3,4,7,8,9,10]. As a consequence, CaPVs are categorized as notifiable diseases under guidelines of the World Organization for Animal Health (OIE) [11].

Next to virus isolation in primary cell cultures [12,13] or continuous cell lines [14], and virus detection via antigen capture ELISAs [15,16] or electron microscopy [17,18], detection of CaPV viral genomes [19] is an important tool for the confirmation of CaPV cases and outbreaks. Here, different PCR detection systems have been developed over the years, ranging from conventional gel-based assays combined with agarose gel electrophoresis [20] or with restriction fragment length polymorphism (RFLP) analyses [21] through loop-mediated isothermal amplification (LAMP) assays [22,23] and high-resolution melting curve analyses [24] to real-time quantitative PCR (qPCR) assays using intercalating dyes [20] or fluorescence-labelled probes [6,25,26].

Conventional PCR assays enable qualitative analysis by agarose gel electrophoresis after PCR amplification [27]. In contrast, real-time PCR assays utilizing intercalating dyes or fluorescence-labelled probes in combination with quenchers allow real-time monitoring of the amplification process [19]. In addition, quantitative PCRs are faster, more specific, and in some cases even more sensitive compared to conventional PCRs [27,28]. Whereas PCR systems using intercalating dyes are combined with melting curve analysis [29,30], probe-based assays do not necessarily need additional working steps [6,25,26]. Nevertheless, all mentioned methods allow various issues to be analyzed reliably: the presence of CaPV genome in general (pan capripox assays) [6,21], the differentiation between the three CaPV species [23,24], and the genetic differentiation of infected from vaccinated animals (DIVA) analyses [20,25,26].

Although many publications exist dealing with several assays for the detection of CaPV viral DNA and further characterization of the respective virus isolates, a probe-based real-time qPCR system for differentiation between lumpy skin disease viruses (LSDV), goatpox viruses (GTPV), and sheeppox viruses (SPPV) is, to our knowledge, still missing.

In the present study, the pan capripox assay of Bowden et al. (2008) was enhanced with an internal control system for control of successful DNA extraction. Furthermore, our already published DIVA assay for LSDV [26] was improved considering recently published LSDV genome sequences. In addition, real-time qPCR assays based on fluorescence-labelled probes were developed that are able to differentiate between all three CaPV species. Finally, several methods of diagnosis and characterization of CaPV were combined with our newly developed species-specific assays to a diagnostic workflow that enables time-saving, reliable, and robust detection, characterization, and genetic examination of CaPV.

## 2. Materials and Methods

Pan capripox real-time qPCR developed by Bowden et al. [6] extended with a modified probe by Dietze et al. [31] served as reference assay for all validation processes. This pan capripox assay was combined with two different control assays during the present study. The first focused on the enhanced green fluorescent protein (EGFP)—DNA as heterologous internal control added during the extraction process [32] and the second used the β-Actin housekeeping DNA [33]. For differentiation of capripox virus species and DIVA strategy of LSDV isolates, probe-based real-time qPCR assays were developed. For analysis of appropriate locations in the genome, several capripox genome sequences were aligned (39 sequences for LSDV field strains, 12 sequences for LSDV vaccine strains, 15 sequences for SPPV strains, and 23 sequences for GTPV strains) using Geneious software package v.11.1.5 (Biomatters, Auckland, New Zealand), and suitable genome regions were selected. Primers and probes finally used are listed in Table 1. The LSDvac-Mix5-Taq-HEX assay, for which sequences of primers and probes are designed on the basis of sequences encoding the Kelch-like protein (ORF008) as described previously [26], turned out to be still suitable due to in-silico analysis during the process of selection of primer and probe sequences. In addition, ORF126 was chosen for design of the LSD-field assay, and sequences of primers and probes of the SPPV-ORF041-MGB-FAM and GTPV- ORF095-Mix1-MGB-HEX assays based on ORF041 and ORF095, respectively.

Each assay can be used as single assay. In addition, the following duplex assays are appropriate: duplex LSDV genetic DIVA assay consisting of LSDfield-ORF126-Mix11-Taq-FAM and LSDvac-Mix5-Taq-HEX and SPPV/GTPV duplex assay consisting of SPPV-ORF041-Mix1-MGB-FAM and GTPV-ORF095-Mix1-MGB-HEX. Validation was performed with the respective duplex assays during the present study. For the LSDV field-specific assay, an alternative LNA probe was tested (LSDfield-LD126-368FAM-LNA; sequence 5′-3′ A(+C)A (+A)C(+G) T(+T)T (+A)T(+G) A(+T)T; brackets indicate the locked nucleic acids).

Primer and probe concentrations were used as presented in Table 2. Initial primer and probe concentrations given by the manufacturer were 100 pmol/µL.

For combination of the pan capripox assay with the internal control assays, two kits were tested. First, the QuantiTect Multiplex-PCR Kit no ROX (Qiagen, Hilden, Germany) with 12.5 μL of reaction mix (1.75 μL of water, 6.25 μL of 2× QuantiTect Multiplex PCR NoROX Master Mix, 1 μL of each primer probe mix, 2.5 μL of DNA) for each sample was used with the following thermal cycling conditions:
15 min95 °C
45 s95 °C
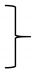

15 s60 °C45 cycles15 s72 °C


Second, the PerfeCTa qPCR ToughMix (Quanta BioSciences, Gaithersburg, MD, USA) with 12.5 µL of reaction mix (10 µL master mix (1.75 μL of water, 6.25 μL of PerfeCTa qPCR ToughMix, 1 μL of each primer probe mix) plus 2.5 μL of DNA) was utilized with the following thermal cycling protocol:
3 min95 °C

15 s95 °C
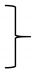

15 s60 °C45 cycles15 s72 °C


Since the PerfeCTa qPCR ToughMix provides a shorter protocol, both newly developed species-specific duplex assays were performed utilizing this kit. However, modified thermal cycling conditions were necessary for the species-specific duplex assays:
3 min95 °C

15 s95 °C
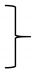

5 s62 °C45 cycles15 s72 °C


Cut-off was set at Cq 40.00 with regard to our experiences with the used reference assay during several studies testing diverse specimen from experimentally infected animals [26,34,35,36,37].

Analytical sensitivity was determined using species-specific standards (based on the LSDV-“Macedonia2016” field strain, LSDV-“Neethling” vaccine strain [26], GTPV-“V/103” field strain, and SPPV-“V/104” vaccine strain [35]) with defined genome copy no./µL that were tested in the pan capripox real-time qPCR, and 15 replicates of each dilution step were analyzed using the specific duplex assays. In addition, analytical specificity was examined using a panel of samples consisting of capripox virus DNA extracted from different sample matrices (e.g., cell-culture as well as EDTA blood, serum, nasal and oral swabs, and different organs of experimentally infected sheep, goats, and cattle, respectively) in duplicates. All samples were previously tested using the pan capripox real-time qPCR and turned out to be positive for capripox viral genome. In addition, the pan capripox assay was used during the same PCR run as the species-specific assays to validate positivity of the samples. Furthermore, all animal specimens were taken in accordance with the regulations, and results of animal specimens are published [26,34,35,36,37].

For further analysis of usability of these assays for examination of field outbreaks, mixed infections of SPPV and GTPV as well as LSDV infections of cattle previously vaccinated with live attenuated vaccine were simulated. Therefore, extracted DNA samples were pooled respectively, and pooled samples were analyzed with the respective duplex assays in duplicates.

## 3. Results

### 3.1. Duplex Assays for Detection of Capripox Virus Genomes and Internal Controls

The previously published pan capripox assay [6,31] was combined with two different internal control (IC) assays, in detail β-Actin [33] and IC-2 DNA (EGFP) that is added during the extraction process [32] using two different qPCR kits. The dilution series of LSDV-“Macedonia2016” was tested using the pan capripox assay with and without internal control assays. Cq-values of both tested duplex assays are highly comparable to those of the single-plex pan capripox assay, independently of the used kit (Table 3).

Conclusively, both duplex assays turned out to be as sensitive as the single-plex pan capripox assay (Table 3), providing a helpful tool during diagnosis of capripox viruses. Furthermore, suitability of both tested kits displays an important redundant system.

### 3.2. Species-Specific Duplex Real-Time qPCR Assays

Since alignment of full-length genome sequences of capripox viruses including recently published genome sequences revealed several nucleotide mismatches of previously published LSDV field-specific probe-based real-time qPCR assays, the “LSDV DIVA 2” assay [26] was improved during the present study. Therefore, a new LSDV field-specific assay was developed based on available ORF126 gene sequences of 51 LSDV strains. Since this assay should also be appropriate in recently vaccinated cattle, it was of marked importance that there be no cross-reactivity with primers and probes of the LSDV field assay with LSDV vaccine genome. After validating both sensitivity and specificity of the single-plex assay (data not shown), both LSDV-specific assays, the LSDV field assay and the LSDV vaccine assay, were tested as duplex real-time qPCRs. Sensitivity of this duplex assay was examined using dilution series of a representative LSDV vaccine strain (LSDV-“Neethling” vaccine) and an already described LSDV field strain (LSDV-“Macedonia2016”). Thereby, both LSDfield-ORF126-Mix11-Taq-FAM and LSDvac-Mix5-HEX turned out to be highly sensitive, both detecting ≤ 10 genome equivalents per µL DNA, and therewith are highly similar to the reference method (Table 4, Appendix A). Furthermore, a redundant LSDV field-specific assay was developed and tested, using the same primers but a different probe type (LNA probe instead of TaqMan probe, see Section 2. Material and Methods), which shows similar results when combined with LSDvac-Mix5-HEX (Appendix A).

Interestingly, the recombinant LSDV strain (Saratov/2017-MH646674.1) was not available for lab testing, but the in-silico analyses of the LSDV assays using Geneious software show clearly that both assays (the LSD-field-ORF126-Mix11-Taq-FAM and the LSDvac-Mix5-Taq-HEX) will not detect this recombinant strain LSDV/Russia/Saratov/2017 (data not shown). From the diagnostic point of view, these results are excellent because the pan-capripox virus p32 assay will react positive and both specific assays for the characterization of LSDV will react negative. Based on these results, first indications for a recombinant strain are given and alternative methods must be used for the further characterization of the strain (e.g., Saratov specific qPCR or sequencing).

The analytic specificity of the duplex assay was tested with a large panel of samples derived from different matrices (e.g., cell culture and samples taken from sheep, goats, and cattle experimentally infected with respective CaPV isolates). Here, again the Cq-values of both LSDV-specific assays showed high similarities to the pan capripox reference assay. In addition, neither cross-reactivity of the LSDV field-specific assay with LSDV vaccine DNA nor false positive results of the LSDV vaccine-specific assay with LSDV field DNA could be observed. Furthermore, as expected, all SPPV and GTPV DNA samples scored negative in both LSDV-specific assays (Table 5).

Finally, the appearance of virulent LSDV in recently vaccinated cattle herds was simulated by mixing DNA from LSDV field strains with LSDV vaccine DNA. These mixed samples were tested with both the single assays and the duplex LSDV-genetic DIVA assay very successfully. Comparison between the single assays and the duplex assay showed highly comparable Cq-values, and no differences in the sensitivity could be observed (Table 6), displaying a great suitability of this duplex LSDV-DIVA assay for analyses in the field.

Next to improvement of the duplex LSDV-DIVA assay, probe-based real-time qPCR assays for species-specific detection of SPPV and GTPV were developed. Therefore, 15 available full-length genome sequences of SPPV and 23 of GTPV were aligned, and suitable genome regions were selected. Finally, the SPPV-specific assay was located in ORF041, whereas for the GTPV-specific assay, ORF095 was chosen. Analytic sensitivity of single-plex assays and the corresponding duplex assay were tested with dilution series of SPPV-“V/104” and GTPV-“V/103”, respectively. When used as duplex assay, both individual assays showed a sensitivity of approximately 10 genome copy numbers per µL DNA, which is similar to the reference method as well as the improved duplex LSDV-DIVA assay (Table 4, Appendix A). Furthermore, analytic specificity was examined with the same panel of CaPV samples as used for the duplex LSDV-DIVA assay. Here, the duplex assay for species-specific detection of SPPV (FAM) and GTPV (HEX) showed excellent specificity. In detail, neither cross-reactivity with other CaPV isolates nor any false negative results could be observed, and the Cq-values were highly comparable to the reference assay (Table 5). Finally, mixed infection of SPPV and GTPV was simulated, and mixed samples consisting of both SPPV DNA and GTPV DNA were tested with the individual assays as well as the duplex SPPV/GTPV assay. Cq-values did not differ between the single use and the duplex assay, showing great suitability of this duplex SPPV/GTPV assay also during suspected mixed infections of sheep and goats (Table 7).

In summary, all developed duplex assays proved to be highly sensitive. Cq-values of the tested samples as well as of the dilution series were comparable to those of the used reference method, the pan capripox real-time qPCR developed by Bowden et al. (2008) (Table 4 and Table 5), which has a limit of detection (LOD) of less than 10 copies/reaction [6]. Both developed duplex assays for LSDVfield/LSDVvac and SPPV/GTPV had a LOD of ≤10 genome copies/µL DNA (Table 4). In addition, all samples of the validation panel were characterized correctly by all new duplex assays (Table 5). Moreover, all assays were able to reliably detect their respective target DNA also in mixed DNA samples (Table 6 and Table 7), therefore providing a helpful tool in the diagnosis of CaPV outbreaks in flocks of sheep and goats or in cattle herds previously vaccinated with live attenuated LSDV vaccine.

### 3.3. Combination of Different Methods to a Diagnostic Workflow from the Detection of Capripox Viral Genomes to the Final Genetic and Phylogenetic Characterization

Together with already published diagnostic methods for further genetic and phylogenetic characterization of CaPV isolates, we developed a workflow for diagnosis and examination of CaPV samples in our lab (Figure 1). In this workflow, we combined the duplex assay consisting of the pan capripox real-time qPCR of Bowden et al. [6] using a modified TaqMan probe [31], and an internal control system, with our newly developed duplex assays for further characterization of the samples, followed by partial sequencing as described by Adedeji et al. [38] for a first rough phylogenetic overview. Subsequently, CaPV isolates are sequenced using a combined approach of next-generation sequencing with the Illumina platform and third generation nanopore sequencing using the MinION platform, which proved to be a very successful method for the generation of high-quality full-length genome sequences of CaPVs before [34,35].

## 4. Discussion

Diverse PCR systems for the discrimination of the three CaPV species have been developed in the past. A first example is gel-based systems combined with RFLP. Heine et al. published an assay for the differentiation of SPPV from LSDV using *EcoR*V [39], and distinction between SPPV and GTPV could be performed using *Hinf*I [40], *EcoR*I, and *Dra*I digestion [41]. Other gel-based assays focus on a species-specific deletion in the genome of SPPV, leading to a shorter PCR fragment without the additional digestion step [42], or target one species specifically [43,44,45]. Another diagnostic PCR method is conventional PCR followed by either a high-resolution melting curve analysis [24] or performance of a fluorescence melting curve analysis [29,30]. Similar tools have been published for the differentiation of field and vaccine strains of capripox viruses (so-called genetic DIVA). In detail, gel-based systems with a different amplicon length analyzed by agarose gel electrophoresis [20], RFLP [46], or high-resolution melting (HRM) curve [24,47,48] as well as a nested PCR assay [49] have been described. What all these assays have in common is the need for an additional processing step following the PCR. An alternative strategy is the addition of fluorescence-labelled probes with dark quenchers, e.g., TaqMan probes. Generally, these assays provide a clearly increased sensitivity [27] and specificity [25] and are faster than conventional PCR assays followed by additional working steps [27]. This type of real-time qPCRs has been successfully utilized for the discrimination of LSDV field strains from LSDV vaccine strains before [25,26,50]. Real-time qPCR assays developed during our study all consist of fluorescence-labelled probes, which makes the assays highly sensitive and specific (Table 4 and Table 5).

In a first approach, the pan capripox real-time qPCR [6,31] was combined with two different internal control systems: β-Actin [33] and the heterologous IC-2-DNA (EGFP) [32]. Comparison of both duplex assays indicated that β-Actin as well as IC-2-DNA are appropriate candidates in combination with the pan capripox assay. Furthermore, differences between the two PCR-kits used in our study could not be observed (Table 3). However, the PerfeCTa qPCR ToughMix needs a shorter thermal cycling protocol than the QuantiTect Multiplex-PCR Kit no ROX, which is why the PerfeCTa qPCR ToughMix was used for the validation of our species-specific assays. Nevertheless, availability of a redundant system like obtained with the QuantiTect Multiplex-PCR Kit no ROX is of importance.

Recently published full-length genome sequences of LSDV field isolates revealed that our previously described LSDvir-Mix4-FAM [26] and the LSwildPr of Agianniotaki et al. [25] show mismatches, which is why we decided to use a new assay to address this issue. LSDfield-ORF126-Mix11, which consists of a TaqMan probe, was combined with already described LSDvac-Mix5-HEX [26]. This improved duplex LSDV-genetic DIVA assay turned out to be highly sensitive displaying a LOD of ≤10 genome equivalents/µL DNA (Table 4). Furthermore, analytical specificity was excellent as no cross-reactions with the respective contrary LSDV DNA as well as SPPV DNA or GTPV DNA could be observed (Table 5). In addition, a second LSDV field-specific assay using the same primers and an LNA probe was successfully tested. This assay is also highly sensitive and specific, even when used as duplex with the LSDvac-Mix5-HEX assay, and displays a redundant assay to the presented duplex LSDV field assay. LNA probes provide a higher thermal stability than TaqMan probes [51], and the stability of duplex of DNA/RNA and LNA probe is marking [52]. Our results also indicate a clearly higher fluorescence of the LNA-based assay compared to the TaqMan probe-based LSDfield-ORF126-Mix11-FAM. However, TaqMan probes combined with a dark quencher are comparatively less expensive than MGB or LNA probes, which makes them more suitable for larger scale screenings or laboratories with less financial means. Next to analysis of samples consisting of either LSDV field DNA or LSDV vaccine DNA, the improved duplex LSDV-DIVA assay is able to reliably detect the respective target DNA in mixed LSDV-DNA samples (Table 6), which plays an important role in areas that vaccinate against LSDV but still suffer from outbreaks. In this case, it is essential that the used duplex assay is able to detect both the LSDV vaccine strain of previously vaccinated cattle and the LSDV field strain that might have affected a vaccinated herd.

Like the duplex LSDV-DIVA assay, the duplex SPPV/GTPV assay shows high analytical sensitivity of approximately 10 genome copies/µL DNA (Table 4), and excellent specificity could be observed during our study (Table 5). Since some isolates of SPPV and GTPV are able to infect both hosts [7], reliable molecular discrimination between SPPV and GTPV is of importance in the field. The described duplex SPPV/GTPV assay therefore provides a great tool for future diagnosis and characterization of SPPV and GTPV in mixed flocks or endemic areas. Moreover, mixed infections of sheep and goats with SPPV and GTPV can also be detected reliably (Table 7). In contrast to the TaqMan-based LSDV-DIVA assay, the SPPV/GTPV assay consists of a minor groove binder (MGB) probe (Table 1). When these DNA probes hybridize with single stranded DNA, an extremely stable duplex is formed, which enables the use of probes that are relatively short [53]. In our cases, the melting temperature of the SPPV and GTPV probes was too low based on a lot of A/T nucleotides. Therefore, the application of TaqMan probes was not possible. Nevertheless, alternative LNA probes for the SPPV and the GTPV assay should be possible, but must be still evaluated carefully.

Another remarkable advantage of the developed duplex assays is their time-saving protocol, which enables fast detection and discrimination between SPPV, GTPV, and LSDV field and vaccine strains. Moreover, the used cycling conditions are also appropriate for the pan capripox reference assay if the pan capripox assay and the species-specific assays shall be performed in one PCR run.

Finally, we combined our newly developed species-specific duplex assays with already published methods for diagnosis and characterization of capripox viruses [6,31,34,35,38] resulting in a complete fit-for-purpose diagnostic workflow (Figure 1). This workflow consists of methods standardly used in our lab. Nevertheless, addition of certain assays or exchange of redundant methods such as the pan capripox assay of Ireland and Binepal (1998) [21] or Haegeman et al. (2013) [54] is possible. Future tests on probe-based real-time qPCR assays for distinction between field and vaccine strains of SPPV and GTPV should be made, and such assays should be implemented into the workflow. Therefore, it has to be noted that development of universal vaccine-specific assays for SPPV and GTPV is challenging. Although possible genes involved in virulence of CaPV have been reported, e.g., ankyrin repeat proteins, kelch-like proteins, and poxvirus B22R superfamily proteins [55,56,57], reliable universal genetic markers for virulence of SPPV and GTPV are still not determined. Nevertheless, development of vaccine-specific assays depending on the used vaccine is possible and may provide a robust DIVA strategy for SPPV and GTPV. Since shedding of vaccine virus DNA seems to be not significant in sheep after vaccination with SPPV vaccine [35], the DIVA strategy might be less important for SPPV and GTPV than for LSDV.

In summary, our diagnostic workflow enables quick, fail-safe, and repeatable classification of capripox virus isolates that is easy to perform by guiding the examiner from extraction of DNA through detection and differentiation of CaPV to final genetic and phylogenetic characterization of CaPV isolates.

## 5. Conclusions

In the present study, we developed duplex probe-based real-time qPCR assays for species-specific differentiation of capripox viruses, namely lumpy skin disease virus, goatpox virus, and sheeppox virus and for discrimination of LSDV field strains and LSDV vaccine strains. To our knowledge, these are the first duplex assays for a reliable distinction of the three capripox virus species that do not need further processing like, for example, melting curve analysis. All assays showed a high analytic sensitivity of approximately 10 genome copy numbers/µL DNA and were highly specific, even for simulated mixed infections of GTPV and SPPV as well as LSDV field infections after recently performed LSDV vaccination.

## Figures and Tables

**Figure 1 microorganisms-09-00765-f001:**
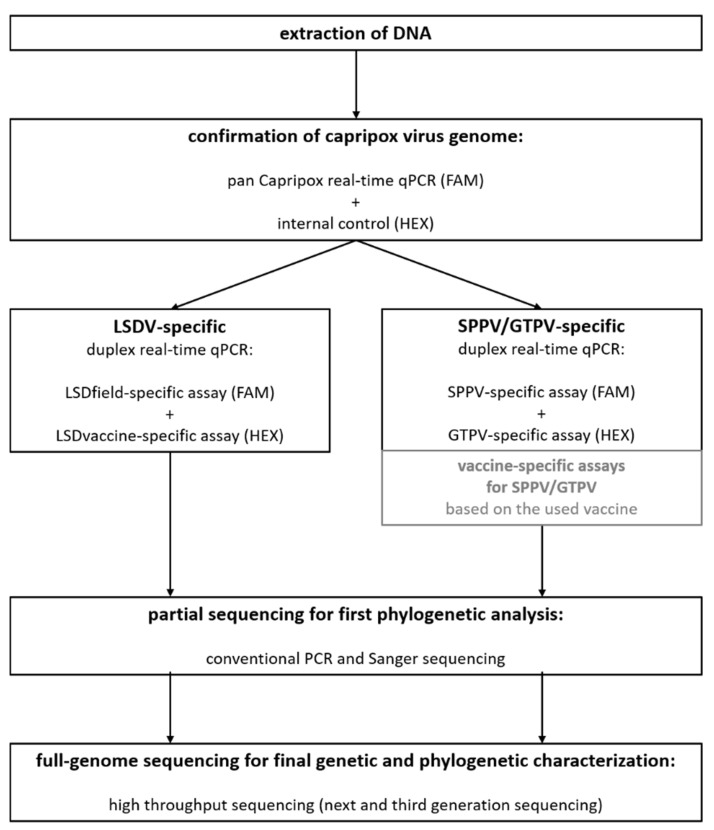
Final workflow for a fit-for-purpose diagnosis and characterization of capripox virus isolates in clinical samples consisting of the detection of viral DNA, the differentiation between the three CaPV species as well as a LSDV-genetic DIVA approach and rough and detailed genetic and phylogenetic characterization. Pan-capripox real-time qPCR combined with an internal control system is followed by the newly developed differentiation assays. Subsequently, partial as well as full-length sequencing is performed. Vaccine-specific assays for SPPV and GTPV (gray boxes) can be included into this workflow but have not been part of our internal lab workflow until now.

**Table 1 microorganisms-09-00765-t001:** Real-time qPCR assays used during the present study for detection of capripox viral genome, internal control and differentiation of the three capripox virus species goatpox virus (GTPV), sheeppox virus (SPPV), and lumpy skin disease virus (LSDV), as well as distinction of LSDV field strains and LSDV vaccine strains. Amplicon length bases on the respective reference sequences for LSDV field strain (NC_003027), SPPV (NC_004002), and GTPV (NC_004003). For LSDV vaccine strain, full-length genome sequence with Accession No. AF409138 was used.

Assay	Designation of Oligo	Sequence of Oligo 5′–3′	Amplicon Length
Capri-p32-Mix1-Taq-FAM [6,31]	Capri-P32for	AAA ACG GTA TAT GGA ATA GAG TTG GAA	89 bp
	Capri-P32-rev	AAA TGA AAC CAA TGG ATG GGA TA	
	Capri-P32-FAM-Taq	FAM-ATG GAT GGC TCA TAG ATT TCC TGA T-BHQ1	
EGFP-Mix 1 (limit5) HEX [32]	EGFP1-F	GAC CAC TAC CAG CAG AAC AC	132 bp
	EGFP2-R	GAA CTC CAG CAG GAC CAT G	
	EGFP-Probe 1	HEX-AGC ACC CAG TCC GCC CTG AGC A-BHQ1	
β-Actin-DNA-Mix2-HEX [33]	ACT2-1030-F	AGC GCA AGT ACT CCG TGT G	96 bp
	ACT-1135-R	CGG ACT CAT CGT ACT CCT GCT T	
	ACT-1081-HEX	HEX-TCGCTGTCCACCTTCCAGCAGATGT-BHQ1	
LSDfield-ORF126-Mix11-Taq-FAM	LSDfield-LD126-341F	GTG AAG AAA ATT TAA TTT GGG AYG A	80 bp
	LSDfield-LD126-420R	GTT AGG TGG TAA ATC ATA AAC ACT A	
	LSDfield-LD126-368FAM	FAM-ACA ACG TTT ATG ATT TAC CRC CTA ATG-BHQ1	
LSDvac-Mix5-Taq-HEX [26]	LSDvac-136790-F	TCT TGG ACA ACT TTG ATG CAT C	127 bp
	LSDvac-136916-R	CTT CAT AGC CTA TTC CGA GAG	
	LSDvac-136891-HEXas	HEX-ACT TGC GTA ACT AAT TCC ACC CAC AA-BHQ1	
SPPV-ORF041-Mix1-MGB-FAM	SPPV-ORF041-53F	AGG TAC AAA ATA ATA CCA ACG ATT C	109 bp
	SPPV-ORF041-161R	GTT GAT TTT TCA ACA TTT ATG TAT TGG	
	SPPV-ORF041-98FAM-MGB	FAM-TGG TAA AAT CAA CAA ATA ATT TTA TTG-MGB	
GTPV-ORF095-Mix1-MGB-HEX	GTPV-ORF095-325F	CAT TTG TTG ATA TAA ACG TTC TTT ACC	140 bp
	GTPV-ORF095-464R	CTA RAG ATT TAG AAA CRA CGG TAA AA	
	GTPV-ORF095-378HEX-MGB	HEX-ATG TAA CAG ATT TGT TTT TAA TT-MGB	

**Table 2 microorganisms-09-00765-t002:** Primer and probe concentrations used for the different real-time qPCR assays. Volumes are given for 100 µL primer-probe mix. Initial concentration of primer and probes were 100 pmol/µL.

Assay	Forward/Reverse Primer	Probe	0.1× TE bufer pH 8.0
Capri-p32-Mix1-Taq-FAM	7.5 µL	2.5 µL	82.5 µL
EGFP-Mix 1 (limit 5) HEX	2.5 µL	2.5 µL	92.5 µL
β-Actin-DNA-Mix2-HEX	2.5 µL	2.5 µL	92.5 µL
LSDfield-ORF126-Mix11-Taq-FAM	7.5 µL	2.5 µL	82.5 µL
LSDvac-Mix5-Taq-HEX	5.0 µL	2.5 µL	87.5 µL
SPPV-ORF041-Mix1-MGB-FAM	7.5 µL	2.5 µL	82.5 µL
GTPV-ORF095-Mix1-MGB-HEX	7.5 µL	2.5 µL	82.5 µL

**Table 3 microorganisms-09-00765-t003:** Cq-values of the pan capripox real-time qPCR (FAM) when combined with either EGFP-Mix1 (limit 5) HEX or β-Actin-DNA-Mix-HEX.

Dilution Step of LSDV-“Macedonia2016”	PerfeCTa qPCR ToughMix			QuantiTect Multiplex-PCR Kit No ROX		
	Cq-Value of			Cq-Value of		
	Pan Capripox Assay (No IC)	Pan Capripox Assay/EGFP Mix 1 (Limit 5) HEX	PAN Capripox Assay/β-Actin-DNA-Mix-HEX	Pan Capripox Assay (No IC)	Pan Capripox Assay/EGFP Mix 1 (Limit 5) HEX	Pan Capripox Assay/β-Actin-DNA-Mix-HEX
10-1	17.96	17.88/29.5	18.01/26.10	18.38	18.29/25.61	18.18/25.10
10-2	21.27	21.25/29.1	21.32/27.34	21.77	21.53/25.65	21.61/26.24
10-3	24.51	24.6/29.00	24.59/27.32	25.2	25.03/26.09	25.04/27.04
10-4	28.02	27.94/29.10	27.84/27.63	28.6	28.33/26.26	28.32/27.41
10-5	31.41	30.91/28.80	30.71/27.74	31.98	31.10/26.14	31.38/27.61
10-6	35.54	33.73/29.04	33.94/27.69	34.5	34.01/26.34	34.25/27.69
10-7	37.37	38.87/28.65	38.04/27.56	no Cq	37.30/26.08	37.58/27.56
10-8	no Cq	no Cq/29.27	no Cq/27.70	no Cq	no Cq/26.21	no Cq/27.63

**Table 4 microorganisms-09-00765-t004:** Sensitivity of the optimized duplex LSDV genetic differentiation of infected from vaccinated animals (DIVA) assay as well as the duplex real-time qPCR assays for differentiation of SPPV and GTPV. The pan capripox real-time qPCR with standard thermal cycling conditions served as reference method.

Genome Equivalents Per µL DNA	Number of Positive Replicates/Number of Overall Replicates				
	Capri-p32-Mix1-Taq-FAM	LSD-field-ORF126-Mix11-Taq-FAM	LSDvac-Mix5-HEX	SPPV-ORF041-Mix1-MGB-FAM	GTPV-ORF095-Mix1-MGB-HEX
10^4^	7/7	15/15	15/15	15/15	15/15
10^3^	7/7	15/15	15/15	15/15	15/15
10^2^	7/7	15/15	15/15	15/15	15/15
10^1^	7/7	15/15	15/15	15/15	15/15
10^0^	7/7	15/15	11/15	13/15	15/15

**Table 5 microorganisms-09-00765-t005:** Specificity of the newly developed duplex assays for the discrimination of different CaPV species and for differentiation of LSDV field and LSDV vaccine strains. All samples were tested in duplicates. Capri-p32-Mix1-Taq-FAM (Capri-p32) served as reference assay. LSD-field-ORF126-Mix11-Taq-FAM (LSDfield) was tested as duplex assay in combination with LSDvac-Mix5-HEX (LSDvac). Moreover, SPPV-ORF041-Mix1-MGB-FAM (SPPV) and GTPV-ORF095-Mix1-MGB-HEX (GTPV) were tested as duplex assays. The mean Cq-values are presented, and the cut-off was set at Cq 40.0.

Sample		Sample Matrix [Reference]	Capri-p32	LSDfield	LSDvac	SPPV	GTPV
LSDV field	V/96	cell culture	24.7	26.1	no Cq	no Cq	no Cq
	V/101	cell culture	14.7	16.1	no Cq	no Cq	no Cq
	V/107	cell culture	15.3	17.2	no Cq	no Cq	no Cq
	V/281	cell culture	13.9	15.4	no Cq	no Cq	no Cq
	BH 50/19-07	proficiency test sample EU 2019	35.8	37.1 *	no Cq	no Cq	no Cq
	BH 24/20-13	proficiency test sample EU 2020	28.0	29.4	no Cq	no Cq	no Cq
	BH 24/20-17	proficiency test sample EU 2020	33.6	34.8	no Cq	no Cq	no Cq
	R/921 EDTA blood 10 dpi	EDTA blood [36]	28.3	30.1	no Cq	no Cq	no Cq
	R/276 EDTA blood 10 dpi	EDTA blood [36]	23.9	25.2	no Cq	no Cq	no Cq
	R/988 EDTA blood 10 dpi	EDTA blood [36]	27.6	29.1	no Cq	no Cq	no Cq
	R/988 serum 11 dpi	serum [36]	27.9	28.9	no Cq	no Cq	no Cq
	R/792 serum 9 dpi	serum [37]	31.9	33.0	no Cq	no Cq	no Cq
	R/280 nasal swab 13 dpi	nasal swab [36]	30.0	31.3	no Cq	no Cq	no Cq
	R/981 nasal swab 11 dpi	nasal swab [36]	23.6	25.0	no Cq	no Cq	no Cq
	R/988 nasal swab 11 dpi	nasal swab [36]	22.9	24.1	no Cq	no Cq	no Cq
LSDV vaccine	V/100	cell culture	14.4	no Cq	13.5	no Cq	no Cq
	V/102	cell culture	18.3	no Cq	17.8	no Cq	no Cq
	V/106	cell culture	14.1	no Cq	13.3	no Cq	no Cq
	V/122	cell culture	16.5	no Cq	15.7	no Cq	no Cq
	V/126	cell culture	15.5	no Cq	14.8	no Cq	no Cq
	BH 50/19-03	proficiency test sample EU 2019	29.9	no Cq	28.8	no Cq	no Cq
	BH 50/19-06	proficiency test sample EU 2019	27.4	no Cq	26.4	no Cq	no Cq
	BH 24/20-15	proficiency test sample EU 2020	30.4	no Cq	29.6	no Cq	no Cq
	R/129 nasal swab 7 dpi	nasal swab [26]	33.1	no Cq	32.1	no Cq	no Cq
SPPV	V/104	cell culture	14.8	no Cq	no Cq	15.7	no Cq
	V/123	cell culture	17.6	no Cq	no Cq	18.4	no Cq
	V/293	cell culture	18.3	no Cq	no Cq	19.2	no Cq
	BH 50/19-01	proficiency test sample EU 2019	27.1	no Cq	no Cq	27.9	no Cq
	BH 24/20-18	proficiency test sample EU 2020	25.6	no Cq	no Cq	26.2	no Cq
	S-02 EDTA blood 14 dpi	EDTA blood [34]	27.0	no Cq	no Cq	28.3	no Cq
	S-09 EDTA blood 12 dpi	EDTA blood [34]	28.9	no Cq	no Cq	30.0	no Cq
	S-13 EDTA blood 10 dpi	EDTA blood [34]	26.9	no Cq	no Cq	28.1	no Cq
	S-02 serum 14 dpi	serum [34]	30.4	no Cq	no Cq	33.0	no Cq
	S-13 serum 10 dpi	serum [34]	33.1	no Cq	no Cq	34.6	no Cq
	S-05 nasal swab 12 dpi	nasal swab [34]	14.1	no Cq	no Cq	15.1	no Cq
	S-12 nasal swab 14 dpi	nasal swab [34]	18.9	no Cq	no Cq	19.6	no Cq
	S-06 oral swab 12 dpi	oral swab [34]	24.1	no Cq	no Cq	25.1	no Cq
	S-12 oral swab 10 dpi	oral swab [34]	27.3	no Cq	no Cq	28.3	no Cq
	S-11 lung	organ sample [34]	23.1	no Cq	no Cq	23.8	no Cq
	S-15 skin lesion prepuce	organ sample [34]	16.5	no Cq	no Cq	17.6	no Cq
	S-04 nasal septum	organ sample [34]	20.4	no Cq	no Cq	21.5	no Cq
	S-03 crust	organ sample [34]	13.9	no Cq	no Cq	14.8	no Cq
GTPV	V/103	cell culture	16.2	no Cq	no Cq	no Cq	16.8
	V/279	cell culture	23.9	no Cq	no Cq	no Cq	24.4
	BH 24/20-11	proficiency test sample EU 2020	31.7	no Cq	no Cq	no Cq	33.4
	BH 24/20-12	proficiency test sample EU 2020	34.9	no Cq	no Cq	no Cq	36.6
	Z/254 EDTA blood 7 dpi	EDTA blood [35]	26.1	no Cq	no Cq	no Cq	27.4
	Z/254 EDTA blood 10 dpi	EDTA blood [35]	24.7	no Cq	no Cq	no Cq	25.9
	Z/256 EDTA blood 13 dpi	EDTA blood [35]	28.1	no Cq	no Cq	no Cq	29.0
	Z/254 serum 10 dpi	serum [35]	34.7	no Cq	no Cq	no Cq	37.7
	Z/256 serum 13 dpi	serum [35]	36.3	no Cq	no Cq	no Cq	36.9
	Z/259 serum 23 dpi	serum [35]	30.6	no Cq	no Cq	no Cq	32.4
	Z/253 nasal swab 10 dpi	nasal swab [35]	25.7	no Cq	no Cq	no Cq	27.0
	Z/256 nasal swab 10 dpi	nasal swab [35]	18.5	no Cq	no Cq	no Cq	19.3
	Z/257 nasal swab 13 dpi	nasal swab [35]	18.0	no Cq	no Cq	no Cq	18.6
	Z/259 nasal swab 21 dpi	nasal swab [35]	23.3	no Cq	no Cq	no Cq	24.0
	Z/256 oral swab 15 dpi	oral swab [35]	24.4	no Cq	no Cq	no Cq	25.1
	Z/257 oral swab 13 dpi	oral swab [35]	28.3	no Cq	no Cq	no Cq	29.4
	Z/259 cervical lymph node	organ sample [35]	20.2	no Cq	no Cq	no Cq	22.4
	Z/255 lung	organ sample [35]	23.8	no Cq	no Cq	no Cq	25.3
	Z/260 skin chest	organ sample [35]	18.8	no Cq	no Cq	no Cq	19.4
	Z/254 skin nose	organ sample [35]	20.7	no Cq	no Cq	no Cq	22.0
	Z/253 trachea	organ sample [35]	19.8	no Cq	no Cq	no Cq	20.4

* Displays samples that were positive in only one out of two PCR runs.

**Table 6 microorganisms-09-00765-t006:** Suitability of duplex assay consisting of LSD-field-ORF126-Mix11-Taq-FAM and LSDvac-Mix5-HEX detecting virulent LSDV strains in cattle herds previously vaccinated against LSDV. First, DNA samples with similar Cq-values were combined. Then, a 1:100 dilution of one of the two samples (dil. 1:100) was prepared and mixed with the respective undiluted DNA sample (undil.).

Sample	Capri-p32	LSD-Field-ORF126-Mix11-Taq-FAM		LSDvac-Mix5-HEX	
		Single	Duplex	Single	Duplex
V101 undil. + V100 undil.	22.5	24.6	24.2	22.2	22.2
V101 dil. 1:100 + V100 undil.	23.6	31.2	30.9	22.1	22.1
V101 undil. + V100 dil. 1:100	23.4	24.2	24.1	28.9	28.3
V107 undil. + V106 undil.	23.3	25.1	24.9	23.0	23.1
V107 dil. 1:100 + V106 undil.	24.4	31.9	31.5	23.1	23.1
V107 undil. + V106 dil. 1:100	24.2	25.2	24.9	30.2	29.4
V281 undil. + V122 undil.	21.1	23.6	23.4	20.6	20.7
V281 dil. 1:100 + V122 undil.	21.8	30.3	30.6	20.6	20.7
V281 undil. + V122 dil. 1:100	22.3	23.4	23.5	27.7	27.4
V96 undil. + V102 undil.	24.8	26.2	26.3	25.2	25.1
V96 dil. 1:100 + V102 undil.	26.3	32.8	33.3	25.2	25.2
V96 undil. + V102 dil. 1:100	25.3	26.1	26.4	32.0	31.5

**Table 7 microorganisms-09-00765-t007:** Suitability of duplex assay consisting of SPPV-ORF041-Mix1-MGB-FAM and GTPV-ORF095-Mix1-MGB-HEX. Dilution of one of the two samples (dil. 1:100) was prepared and mixed with the respective undiluted DNA sample (undil.).

Sample	Capri-p32	SPPV-ORF041-Mix1-MGB-FAM		GTPV-ORF095-Mix1-MGB-HEX	
		Single	Duplex	Single	Duplex
V103 undil. + V104 undil.	21.6	24.8	24.9	24.3	24.6
V103 dil. 1:100 + V104 undil.	23.3	24.6	24.8	31.4	34.3
V103 undil. + V104 dil. 1:100	22.1	31.6	31.6	24.2	24.3
V279 undil. + V123 undil.	24.6	27.4	27.3	26.8	26.6
V279 dil. 1:100 + V123 undil.	26.1	27.3	27.4	33.6	35.7
V279 undil. + V123 dil. 1:100	25.2	34.5	34.2	26.4	26.3

## Data Availability

The data presented in this study are available in the present manuscript.

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
