# Peer review of "Probe-Based Real-Time qPCR Assays for a Reliable Differentiation of Capripox Virus Species"

_microorganisms, 2021, doi:10.3390/microorganisms9040765_

Round 1

Reviewer 1 Report

Dear Authors,

The presented manuscript is of great value in diagnosing capripox virus infections and the distinction of vaccinal and field strains. My only concern is possible false positive/negative results with recombinant LSDV strains (e.g. LSDV/Russia/Saratov/2017 MH646674.1). If possible, you should test samples of recombinant viruses with your primers. If samples are not available, comparing primers and probes described in this study with available sequences would help address this issue. 

Author Response

The presented manuscript is of great value in diagnosing capripox virus infections and the distinction of vaccinal and field strains. My only concern is possible false positive/negative results with recombinant LSDV strains (e.g. LSDV/Russia/Saratov/2017 MH646674.1). If possible, you should test samples of recombinant viruses with your primers. If samples are not available, comparing primers and probes described in this study with available sequences would help address this issue.

→ We thank the reviewer for this excellentcomment and the benevolent evaluation of our manuscript. Sadly, no recombinant LSDV from Russia were available for testing theLSDV assays.Nevertheless, the in-silicotesting of the LSDV assays shows clearly, that both specific assays (the LSD-field-ORF126-Mix11-Taq-FAMand the LSDvac-Mix5-Taq-HEX) will not detect this recombinantstrain LSDV/Russia/Saratov/2017 MH646674.1. From the diagnostic point of view this resultsareexcellent, becausethe pan-capripox virusp32assay will reactpositiveand thespecific assaysfor the characterisation of LSDVwill react negative. Based on theseresults, first indications for arecombinant strainare given and alternative methods must be used for the further characterisation of the strain (e.g. Saratov specific qPCR or sequencing).

The according information was added to the manuscript.

Reviewer 2 Report

Authors developed several assays for differentiation of capripox viruses. The study is interesting and important but the authors have to improve the manuscript. The paper is a bit hard to follow, it is not transparent. A detailed description of the validation and material used in validation is missing (this is very good described in paper Agianniotaki et al., 2017)

Line 66-70 Please rephrase

Line75-80 Please rephrase

Line 87- 89 These assays can be used as duplex?

Line 93-95 These sentence should be removed from materials and methods section

Line 117-119 These sentences should be removed from materials and methods section. In this section authors should describe only materials and methods. How the cut-off was determined?

Line124-133 Please describe your samples in more detail (origin, if there are positive or negative, whether they have been tested before and what kind of test was used to confirm the presence of infection, etc.).

Results 3.1.  All this chapter is incomprehensible to me. What do the cq values indicate (the virus is detected or internal controls)? If only virus is detected then why are internal controls used? cq should be for virus and for internal controls but it is not. I don't understand this. Please explain.

Results Section 3.2

Generally I don’t understand what the authors mean by “duplex assay”. For me a duplex is when in one reaction you can detect genetic material from two sources (e.g. two viruses or two different virus fragments). Here I only see single assays. Please explain.

 Line 167- no results shown

Line 222-232. These sentences should be removed from results section. This is conclusion.

Author Response

Authors developed several assays for differentiation of capripox viruses. The study is interesting and important but the authors have to improve the manuscript. The paper is a bit hard to follow, it is not transparent. A detailed description of the validation and material used in validation is missing (this is very good described in paper Agianniotaki et al., 2017) 
→ We thank the reviewer for evaluation of our manuscript. All points are answered by a point-topoint reply with the assessment of Reviewer 2 taken also into consideration. 
Line 66-70 Please rephrase 
→ Since Reviewer 2 assessed the manuscript as a “well-written and organised manuscript presenting in-depth validation study”, and the comment of Reviewer 1 is very unspecific and without suggestion for improvement, we would like to stay with the original text.  
Line75-80 Please rephrase 
→ Since Reviewer 2 assessed the manuscript as a “well-written and organised manuscript presenting in-depth validation study”, and the comment of Reviewer 1 is very unspecific and without suggestion for improvement, we would like to stay with the original text.  
Line 87- 89 These assays can be used as duplex? 
→ Each described assay can be used as single-plex assay. In addition, the following duplex assays are suitable:  
- LSDV field/vaccine duplex assay consisting of LSDfield-ORF126-Mix11-Taq-FAM and LSDvacMix5-Taq-HEX - SPPV/GTPV duplex assay consisting of SPPV-ORF041-Mix1-MGB-FAM and GTPV-ORF095Mix1-MGB-HEX 
We clarified this in the manuscript. 
Line 93-95 These sentence should be removed from materials and methods section 
→ This part was removed from the materials & methods and included into discussion. 
Line 117-119 These sentences should be removed from materials and methods section. In this section authors should describe only materials and methods. How the cut-off was determined? 
→ The sentences was included into the discussion instead of materials & methods. In addition, determination of cut-off due to our experiences with animal specimen with the used reference method is explained in the manuscript now. 
Line124-133 Please describe your samples in more detail (origin, if there are positive or negative, whether they have been tested before and what kind of test was used to confirm the presence of infection, etc.). 
→ All samples were previously tested positive using the pan capripox real-time qPCR. In addition, the pan capripox assay was used during the same PCR run as the species-specific assays to validate positivity of the samples. Furthermore, results of animal specimen are still published. This is mentioned in the text now. 
Results 3.1.  All this chapter is incomprehensible to me. What do the cq values indicate (the virus is detected or internal controls)? If only virus is detected then why are internal controls used? cq should be for virus and for internal controls but it is not. I don't understand this. Please explain. 
→ The Cq-values display the Cq-values of the pan capripox assay, what is still described in table 3. Since the aim of this study was not to prove that the internal control systems work (this is still published) but to show that there is no loss of sensitivity of the pan capripox assay when combined with the internal control assays, only Cq-values of the pan capripox assay are shown. 
Results Section 3.2 
Generally I don’t understand what the authors mean by “duplex assay”. For me a duplex is when in one reaction you can detect genetic material from two sources (e.g. two viruses or two different virus fragments). Here I only see single assays. Please explain. 
→ This is described in the material and methods section in more detail now. Each duplex assay detects either capripox virus DNA in addition to internal control DNA (3.1), two different capripox virus species (SPPV and GTPV) or both LSDV vaccine strain DNA and LSDV field strain DNA. Thus, genetic material of two different sources will be detected. 
 Line 167- no results shown 
→ Since the results are highly similar to those of the described LSDV DIVA assay, the results of the duplex assay consisting of the LNA-Mix are shown only in the supplements. Respective supplemental tables (S2-S4) were added. 
Line 222-232. These sentences should be removed from results section. This is conclusion. 
→ We agree with the reviewer that this section was unclear introduced. The named lines contain a summary of the chapter “species-specific duplex real-time qPCR assays” for giving an overview and thus we changed the introducing words for this section. In addition, Reviewer 2 confirmed the structure of the manuscript as “well-written and organised” and thus we would like to stay with nearly the original text. We very much hope that reviewer 1 can accept our statement in this point. 

Reviewer 3 Report

This is a well-written and organised manuscript presenting in-depth validation study on real-time PCR diagnosis of capripox viruses. In view of my expertise it may be accepted in its present form.

Author Response

This is a well-written and organised manuscript presenting in-depth validation study on real-time PCR diagnosis of capripox viruses. In view of my expertise it may be accepted in its present form.

→ We kindly thank the reviewer for this evaluation of our manuscript and work.

Round 2

Reviewer 2 Report

The authors improved the manuscript, however they did not meet all remarks.

Line 67-70  “Pan capripox real-time qPCR developed by Bowden et al. [6] extended with a modified probe by Dietze et al. [31] served as reference assay, which was combined with two different control assays during the present study: EGFP as heterologous internal control added during the extraction process [32] and β-Actin [33], respectively”.  I still stand by my opinion that the sentence is incomprehensible. What does the word respectively refer to?

Line 76-77 “LSDvac-Mix5-Taq-HEX, which is located the Kelch-like protein  encoding gene (ORF008) as described previously [26], turned out to be still suitable.” Authors should be more precise in how they write. LSDvac-Mix5-Taq-HEX is name of the assay so the authors cant write that “ LSDvac-Mix5-Taq-HEX, which is located the Kelch-like protein  encoding gene (ORF008)”. The sequences of primers and probe have been designed on the basis of sequences encoding Kelch-like protein. The second part of the sentence “turned out to be still suitable” should not be included in M/M section-indicates an outcome or conclusion.

Line 77-80 “In 77 addition, ORF126 was chosen for LSD-field assay, and primers and probes of SPPV-78 ORF041-MGB-FAM and GTPV- ORF095-Mix1-MGB-HEX are located in ORF041 and ORF095, respectively”- similar comment as above. Primers and probes are not located… sequences of primers and sequences of probe have been designed on the basis….. The authors should be more precise. What program was used to design?

I still maintain that the description of the samples used to estimate specificity is insufficient. It is not indicated how the authors obtained the samples, whether they were obtained in accordance with the regulations.

Not “All samples were previously tested positive…” but “all samples were previously tested using real-time qPCR and were positive”. I understand that some of the samples used for validation were described in earlier publications that are cited (5 publications) but the authors should briefly characterize these samples. Especially that some, as the authors point out, have not been described anywhere “Furthermore, results of animal specimen are still published”. Moreover, in Table 5, citations should be inserted so that readers can find the original description of the samples.

Section 3.1

The creation of duplexes that do not confirm the possibility of simultaneous detection of a virus and an internal control is meaningless. These results do not confirm that duplex is working properly.

Line 172-173 “After validating both sensitivity and specificity of the  single-plex assay…” no single assays validation results shown or citation

Line 184-192 The description and result are missing. If they are not shown it is suggested to write (data not shown).

304-305 “Comparison of both duplex assays indicated that β-Actin as well as IC-2-DNA are 304 appropriate candidates in combination with the pan capripox assay” there are no results showing that simultaneous virus detection and internal inspection works at all. There is no cq value for the internal controls.

Author Response

Line 67-70 “Pan capripoxreal-time qPCR developed by Bowden et al. [6] extended with a modified probe by Dietze et al. [31] served as reference assay, which was combined with two different control assays during the present study: EGFP as heterologous internal control added duringthe extraction process [32] and β-Actin [33], respectively”. I still stand by my opinion that the sentence is incomprehensible. What does the word respectively refer to?

We tried to change this to a more easy writing:Pan capripox real-time qPCR developed by Bowden et al. [6] extended with a modified probe by Dietze et al. [31] served as reference assay for all validation processes. This pan capripox assay was combined with two different control assays during the present study. The first focused on the enhancedgreen fluorescent protein (EGFP) –DNA as heterologous internal control added during the extraction process [32] and the second used the β-ActinhousekeepingDNA [33].

Line 76-77 “LSDvac-Mix5-Taq-HEX, which is located the Kelch-like protein encoding gene (ORF008) as described previously [26], turned out to be still suitable.” Authors should be more precise in how they write. LSDvac-Mix5-Taq-HEX is name of the assay so the authors cant write that “ LSDvac-Mix5-Taq-HEX, which is located the Kelch-like protein encoding gene (ORF008)”. The sequences of primers and probe have been designed on the basis of sequences encoding Kelch-like protein. The second part of the sentence “turned out to be still suitable” should not be included in M/M section-indicates an outcome or conclusion.

We thank the reviewer for clarification regarding the phrase dealing with the sequences and changed the manuscript accordingly. The part “turned out to be still suitable” refers to the analysis, when we were searching for appropriate primer and probe sequences and not to the results of the validation. We specified this in the text.

Line 77-80 “In 77 addition, ORF126 was chosen for LSD-field assay, and primers and probes of SPPV-78 ORF041-MGB-FAM and GTPV-ORF095-Mix1-MGB-HEX are located in ORF041 and ORF095, respectively”-similar comment as above. Primers and probes are not located... sequences of primers and sequences of probe have been designed on the basis..... The authors should be more precise. What program was used to design?

We thank the reviewer for clarification regarding the phrase dealing with the sequences andchanged this in the manuscript accordingly.Used software is also added.

I still maintain that the description of the samples used to estimate specificity is insufficient. It is not indicated how the authors obtained the samples, whether they were obtained in accordance with the regulations.

As it can be read in the publications that were given at reference, all animal specimens were taken in accordance with the regulations. However, we added this information in the manuscript.

Not “All samples were previously tested positive...” but “all samples were previously tested using real-time qPCR and were positive”. I understand that some of the samples used for validation were described in earlier publications that are cited (5 publications) but the authors should briefly characterize these samples. Especially that some, as the authors point out, have not been described anywhere “Furthermore, results of animal specimen are still published”. Moreover, in Table 5, citations should be inserted so that readers can find the original description of the samples.

We thank the reviewer for the help with the phrase and changed it in the text accordingly.

All animal samples were previously published in the five publications cited. The only samples of which the results werenot published previously are cell culture viruses that were isolated from field outbreaks and propagated in our lab for validation of the assays.Positivity was also confirmed by the pan capripox assay as mentioned in the manuscript.Moreover, proficiencytest samples were not published, but positivity of the samples is confirmed by the EU reference lab for capripox viruses, which was the responsible lab for these tests.

We added the references for each animal sample in the table.

Section 3.1

The creation of duplexes that do not confirm the possibility of simultaneous detection of a virus and an internal control is meaningless. These results do not confirm that duplex is working properly.

We thank the reviewer for this comment and added the respective Cq-values in the table to confirm that the duplex assays work properly.

Line 172-173 “After validating both sensitivity and specificity of the single-plex assay...” no single assays validation results shown or citation

The respective data are not shown in the manuscript as the aim of the manuscript was validation of the duplex assays since they are cost-efficient and can be used in a broader range. “Data not shown” was added into the manuscript.

Line 184-192 The description and result are missing. If they are not shown it is suggested to write (data not shown).

This was added in the manuscript.

304-305 “Comparison of both duplex assays indicated that β-Actin as well as IC-2-DNA are 304 appropriate candidates in combination with the pan capripox assay” there are no results showing that simultaneous virus detection and internal inspection works at all. There is no cq value for the internal controls.

We thank the reviewer for this comment and added the respective Cq-valuesin the table to confirm that the duplex assays work properly.